Age-related changes of dental pulp tissue after experimental tooth movement in rats

Von Böhl Martina 1
Ren Yijin 2
Kuijpers-Jagtman Anne M. 1
Fudalej Piotr S. 3 4 piotr.fudalej@zmk.unibe.ch
Maltha Jaap C. 1
1 Department of Orthodontics and Craniofacial Biology, Radboud University Nijmegen , Nijmegen , The Netherlands
2 Department of Orthodontics, University of Groningen, University Medical Centre Groningen , Groningen, Griningen , The Netherlands
3 Department of Orthodontics and Dentofacial Orthopedics, University of Bern , Bern , Switzerland
4 Department of Orthodontics, Institute of Dental Science, Palacký University Olomouc , Olomouc , Czech Republic
Elangovan Satheesh
Electronic publication date: 2016 Jan 25
Publication date: 2016
Volume: 4
Electronic Location ID: e1625
Received 2015 Aug 10; Accepted 2016 Jan 5
Copyright: ©2016 Von Böhl et al.
Copyright year: 2016
Copyright holder: Von Böhl et al.
License: This is an open access article distributed under the terms of the Creative Commons Attribution License, which permits unrestricted use, distribution, reproduction and adaptation in any medium and for any purpose provided that it is properly attributed. For attribution, the original author(s), title, publication source (PeerJ) and either DOI or URL of the article must be cited.
License URL: https://creativecommons.org/licenses/by/4.0/

Keywords: Dental pulp, Orthodontics, Wear, Ageing, Rats, Tooth movement

Funding: The authors received no funding for this work.

==============================
It is generally accepted that the effect of orthodontic tooth movement on the dental pulp in adolescents is reversible and that it has no long-lasting effect on pulpal physiology. However, it is not clear yet if the same conclusion is also valid for adult subjects. Thus, in two groups of rats, aged 6 and 40 weeks respectively, 3 molars at one side of the maxilla were moved together in a mesial direction with a standardized orthodontic appliance delivering a force of 10 cN. The contralateral side served as a control. Parasagittal histological sections were prepared after tooth movement for 1, 2, 4, 8, and 12 weeks. The pulp tissue was characterized for the different groups, with special emphasis on cell density, inflammatory cells, vascularity, and odontoblasts. Dimensions of dentin and the pulpal horns was determined and related with the duration of orthodontic force application and age ware evaluated. We found that neither in young nor in adult rats, force application led to long-lasting or irreversible changes in pulpal tissues. Dimensional variables showed significant age-related changes. In conclusion, orthodontic tooth movement per se has no long-lasting or irreversible effect on pulpal tissues, neither in the young nor in the adult animals.

Introduction

Conflicting results have been presented on the putative adverse effects of orthodontic tooth movement on the dental pulp. Some claim permanent damage (Hargreaves, Goodis & Seltzer, 2002), but others find no significant long-lasting effects (Unsterseher et al., 1987). It is well known that the application of orthodontic forces induces the release of proinflammatory cytokines in the dental pulp, leading to a temporal acute inflammation and hyperemia (Raiden et al., 1998). Part of this reaction is an upregulation of IL-1α, IL-1β, IL-3, IL-6, and TNFα (Bletsa, Berggreen & Brudvik, 2006; Yamaguchi et al., 2004) and apoptosis (Perinetti et al., 2004; Perinetti et al., 2005; Shigehara, Matsuzaka & Inoue, 2006; Yamaguchi et al., 2004).

Studies in adult rats have shown that the vascular reaction shows biphasic characteristics. An initial decrease for approximately 30 min is followed by a temporary increase in blood flow for the subsequent 24–72 h (Santamaria et al., 2007; Santamaria et al., 2006). Others, however, reported a reversible increase in the number of blood vessels during the first three to seven days of force application (Abi-Ramia et al., 2010; Nixon et al., 1993; Shigehara, Matsuzaka & Inoue, 2006). However, after 6 weeks the vascularity of the pulp had returned to normal, even in cases with severe root resorption (Tripuwabhrut et al., 2010). Studies on isolated human pulp cells derived from premolars extracted during orthodontic tooth movement have shown that increase in vascularity might be caused by a stimulation of angiogenesis through an upregulation of VEGF, FGF2, PDGF, and TGFβ (Derringer & Linden, 1998; Derringer & Linden, 2003; Derringer & Linden, 2004).

Such temporal changes in pulpal blood flow are not only found during tipping movements in rats, but also during intrusion and extrusion in human adults (Barwick & Ramsay, 1996; Sano et al., 2002) and adolescents (Raiden et al., 1998; Ramazanzadeh et al., 2009; Subay et al., 2001).

Temporal vacuolization of the odontoblasts or disruption of the odontoblastic layer in the dental pulp is often described in adolescent humans after intrusion or extrusion (Ramazanzadeh et al., 2009; Stenvik & Mjor, 1971) and in rats and dogs during orthodontic tooth movement (Abi-Ramia et al., 2010; Anstendig & Kronman, 1972) or during intrusion (Abi-Ramia et al., 2010; Konno et al., 2007; Santamaria et al., 2007) . However, others reported no effects in rats during tipping movement (Abi-Ramia et al., 2010; Santamaria et al., 2007).

The above mentioned studies have been performed in adolescent humans or in young adult animals. They suggest that pulpal reactions, evoked by orthodontic interventions are reversible and have no long-lasting effect on pulpal physiology. However, an ever-growing number of adult and elderly individuals seek orthodontic treatment and the question arises whether pulpal reactions on orthodontic therapy change with age.

It is well known that canal and chamber volume is inversely proportional to age. Reparative dentin resulting from restorative procedures, trauma, attrition, and recurrent caries also contributes to decrease of canal and chamber size (Hargreaves & Cohen, 2011). Other age-related changes in the pulp are compromised circulation and innervation, fat droplet deposition, odontoblastic vacuolization, reticular atrophy, pulpal fibrosis, hyaline degeneration, mucoid degeneration, and diffuse calcification (Bernick & Nedelman, 1975; Morse, 1991).

As many of the age-related changes of the dental pulp are also described to be induced by orthodontic interventions, discrimination between the two is difficult (Hargreaves & Cohen, 2011; Hargreaves, Goodis & Seltzer, 2002). Therefore, the present study was meant to investigate the putative differences between rat pulp tissue in young and adult rats and possible additional long-lasting effect of experimental tooth movement, using a standardized appliance exerting a force of constant magnitude and direction. In a previous study this appliance has been shown to result in an almost bodily tooth movement. The hypothesis to be tested was that pulpal effects of orthodontic tooth movement are not age-dependent.

Material and Methods

Ethical permission for the study was obtained according to the guidelines of the Board for Animal Experiments of the Radboud University Nijmegen.

Two groups of 30 male Wistar rats, one aged 6 weeks (150–250 g) and the other 40 weeks (400–550 g) respectively, were acclimatized for 2 weeks before the start of the experiment. They were housed under normal laboratory conditions and fed powdered rat chow (Sniff, Soest, The Netherlands) and water ad libitum. A standard 12-hour light-dark cycle was maintained. The age of 6 weeks in rats corresponds with a circumpubertal age in humans (i.e., 12–15 years), and the age of 40 weeks corresponds with adult age in humans (Sengupta, 2013).

A split-mouth design was used with the experimental side randomly chosen and the contralateral side as the control. Power analysis with the following assumptions: α = 0.05, β = 0.2, d = 5, and s = 3, demonstrated that N = 6 per group (i.e., 6 experimental and 6 control sides for each time point) is sufficient for the analyses with a power of 0.8.

An orthodontic appliance was placed only on the experimental side under general anesthesia (FFM-mix 2.8 ml/kg intraperitoneally, containing fluanisone 6.8 mg/kg, fentanyl 0.1 mg/kg, and midazolam 3.4 mg/kg.) (Janssen, Beerse, Belgium). The appliance has been described extensively elsewhere (Ren, Maltha & Kuijpers-Jagtman, 2004; Ren et al., 2003). In brief, a transverse hole was drilled through the middle root level of the upper incisors and alveolar bone and a stainless steel ligature wire was put through it. A preformed ligature wire (Φ 0.20 mm) enclosing all three upper molars was bonded (Clearfil SE Bond, Kuraray Europe GmbH, Dusseldorf, Germany) to the buccal side of the molars. By this the three molars could be moved mesially as one unit by a Sentalloy® closed coil spring (10 cN, Φ 0.22 mm, eyelet Φ 0.56 mm, GAC, NY, USA) that was attached to the ligature through the upper incisors at one end and the ligature around the three molars at the other end (Fig. 1). This appliance has two major advantages. First, the molar unit is almost bodily moved related to the relative large mesiodistal dimensions of the molar block that is to be moved, and second, the force per molar is comparable to normally used forces in the clinical situation (Ren, Maltha & Kuijpers-Jagtman, 2004; Ren et al., 2003) . After a force application for 1, 2, 4, 8, or 12 weeks, 6 rats from the “young” and 6 animals of the “old” groups were killed.

Figure 1 Schematic drawing of the orthodontic appliance.

The Sentalloy spring delivers a continuous force of 10 cN on all three molars together.

Histological procedure

The rats received a lethal dose of anesthetic before they were killed. They were then perfused with 4% paraformaldehyde solution in 0.1 M PBS at 37 °C. The maxillae were dissected and immersed in 4% paraformaldehyde for 24 h at 4 °C, then they were rinsed in 0.1 M phosphate buffered saline (PBS). After decalcification in 10% ethylenediaminetetraacetate (EDTA) and paraffin embedding, serial parasagittal 7 µm sections were cut. Every 25th section was collected and stained with hematoxylin and eosin.

Measurements

Tooth positions were measured at the control and the experimental sides as the distance between the most mesial point of the first molar and the enamel cementum border of the ipsilateral incisor at the gingival gingival level (I-M distance) at the following time points 0, 1, 2, 4, 8, and 12 weeks. The experimental tooth movement was calculated as the changes in the difference between the I-M distances at the experimental and the conrol side. By this method, possible confounding factors as anchorage loss from the incisors and physiological distal drift of the molars is compensated for.

From each rat three sections were selected from both the experimental and the control side, in which at least one root was completely present including the pulp chamber and the apical foramen. From these sections, the histological features of the dental pulp were described, with special emphasis on odontoblastic layer, blood vessels, cell density, and inflammatory cells in the pulp chamber.

To describe the dimensional changes in tooth morphology during the experimental period in the young as well as in the old group, the following parameters were measured in arbitrary units (AU) as illustrated in Fig. 2: distal and mesial crown height as the distance from the occlusal surface to the cementoenamel junction (CEJ) (1, 2), dentin thickness at the cusps (3, 4, 5), dentin thickness in the fissures (6, 7), predentin thickness at the cusps (8, 9, 10), predentin thickness in the fissures (11, 12), and the height of the pulpal horns, related to a line through the border between dentin and cementum in the fissures (13, 14, 15). Prior to statistical analysis, for each individual side, the means were calculated for the variables crown height, dentin height cusp, dentin height fissure, predentin height cusp, predentin height fissure, and height pulpal horn.

Figure 2 Schematic drawing of a parasagittal section of a rat molar.

The measured parameters are indicated. For explanation see text.

Statistical analysis

The differences between the amount of tooth movement between the young and the old animals was analyzed by Students t tests. The data of the dimensional measurements on the molars from the two age groups were initially analyzed by a two-way ANOVA on the raw data of all variables, with the group (experimental or control), and the duration of force application (1, 2, 4, 8, or 12 weeks) as independent factors. As this analysis showed no significant group effect for any of the variables, the data from the experimental and the control sides were pooled. Subsequently, the time effects were analyzed by one-way ANOVA on all dimensional variables in both age groups separately with the experimental period as independent variable.

A separate one-way ANOVA was performed after the data from both age groups were combined to study the long term changes in these parameters with the real age of the animals as independent factor. A linear regression analysis was performed and the explained variance was calculated for all parameters in the two age groups and in the combined age group.

Results

Experimental tooth movement

Throughout the whole experimental period the teeth in the young animals had moved significantly more than in the old animals. The teeth showed initially a fast movement of 1.05 ± 0.41mm/w for the young and 0.38 ± 0.28 mm/w for the old animals. The rate of tooth movement slowed down to about 0.05 mm/w for the young and 0.02 mm/w for the old animals at the end of the experimental period. The total tooth movement over the experimental period of 12 weeks was 2.69 ± 0.62 mm for the young and 1.23 ± 0.56 mm for the old animals, and the tooth movement curves can best be described by logarithmic equations, showing an R2 of 0.996 and 0.965 for the young and the old group, respectively.

Histological dental pulp tissue survey

Control side in young rats

No histological differences were encountered between animals, which were for a short (1–2 weeks) or long period (8–12 weeks) in the experiment. The pulps in the control teeth showed only very few inflammatory cells and small blood vessels (Fig. 3A). The odontoblastic layer and cementum surface showed an irregular morphology. At the roof of the pulp chamber and in the pulpal horns just a thin predentine layer was seen (Fig. 3A). The tips of the cusps showed very little or no wear (Fig. 3C).

Figure 3 Histological sections of young (A, B, C) and old rats (D, E, F).

The most clear difference between the control and experimental pulpal tissue is that the latter show some more and wider blood vessels. The most prominent difference between the young and old animals is that in the latter, the odontoblasts are more well-organized and active, and the predentin layer is more pronounced. The low-power pictures show the dramatic difference in wear between the young and old animals. All sections H & E staining.

Experimental side in young rats

Again, no histological differences were encountered between animals, which were for a short (1–2 weeks) or a long period (8–12 weeks) in the experiment.

The pulp of the experimental teeth showed a lower cell density than the controls, and inflammatory cells were scarce. The number of blood vessels was increased compared to the controls and in general their diameter was larger (Fig. 3B). The odontoblastic layer and cementum surface were irregular. The predentin layer at the top of the pulp chamber appeared to be slightly thicker than in the controls (Fig. 3B). Similar to the control animals, the tips of the cusps showed very little or no wear.

Control side in old rats

No substantial histological differences were found between animals that stayed for a short (1–2 weeks) or long period (8–12 weeks) in the experiment. In the control teeth from the old group, the pulpal cell density was less than in the young animals. Only few inflammatory cells were present. A limited number of small blood vessels were present throughout the pulp (Fig. 3D).

The odontoblastic layer and predentin were well organized. At the top of the pulp chamber the predentin layer tended to be somewhat thicker than in the young animals (Fig. 3D). The tips of the cusps showed severe wear (Fig. 3F).

Experimental side in old rats

Again, no substantial histological differences were found between animals that stayed for a short (1–2 weeks) or long period (8–12 weeks) in the experiment. The number of inflammatory cells in the dental pulp was larger than in the control teeth. The number of blood vessels was larger and they showed a larger diameter compared to the controls (Fig. 3E). The odontoblastic cell layer and the predentin surface were organized in a regular way. The predentin layer in the pulpal horns and the roof of the pulp chamber was thicker than in the young animals and similar to the control animals in the old group (Fig. 3E). The tips of the cusps showed severe wear, similar to the controls. The amount of wear increased during the experimental period.

Dimensional parameters

Figs. 3C and 3F illustrate the general morphology of the first molars in the young and the old group.

Two-way ANOVA with the group (experimental or control), and duration of force application (1, 2, 4, 8, or 12 weeks) as independent factors showed no significant group effect for any of the variables. Therefore, the data from the experimental and the control groups were pooled (Table 1).

Table 1 Means and standard deviations (sd) for the combined data of the dimensional parameters in the young and the adult groups.

Age (weeks)	Crown	Dentin	Predentin	Pulpal horn	
	Height	Cusp	Fissure	Cusp	Fissure	Height	
	Mean	sd	Mean	sd	Mean	sd	Mean	sd	Mean	sd	Mean	sd	
Young group	
7	33.3	2.5	16.5	2.5	12.4	1.9	0.8	0.2	0.1	0.2	2.5	3.4	
8	30.0	2.5	14.4	2.2	13.1	2.2	0.8	0.3	0.1	0.1	3.1	2.7	
10	29.4	4.0	16.5	1.9	13.3	2.4	1.0	0.3	0.1	0.2	1.5	2.2	
14	31.0	2.7	16.0	2.5	14.3	2.6	1.0	0.1	0.1	0.1	2.7	2.4	
18	33.4	3.7	15.2	1.3	14.7	2.6	0.8	0.2	0.1	0.1	3.2	1.8	
Old group	
41	17.8	3.1	6.2	2.5	14.3	0.8	0.9	0.2	0.4	0.2	−0.3	1.5	
42	18.4	2.7	6.6	2.4	15.2	1.1	0.8	0.3	0.2	0.2	−1.2	1.1	
44	20.6	2.8	7.0	3.0	10.8	2.8	0.7	0.1	0.5	0.3	1.6	2.8	
48	18.2	3.4	8.1	3.3	10.4	4.4	0.8	0.2	0.4	0.4	0.6	2.1	
52	8.3	4.1	13.4	4.8	14.1	2.0	1.1	0.4	0.5	0.3	1.6	1.8	

The one-way ANOVA for none of the mean variables in the young group showed any significant time effect, low correlation coefficients (R ≤ 0.2) and a very small explained variance (R2 ≤ 0.05) (Table 2). In the old group only the mean crown height showed a significant decrease, and the mean thickness of the dentin in the cusps a significant increase over time. The explained variance (R2) was 0.42 and 0.67 respectively (Table 2).

Table 2 Statistical analysis of the effect of experimental period (in the young and the old group) and real age (in the combined data) for the dimensional measurements.

Variable	Young	Old	Combined	
	ANOVA	Linear regression	ANOVA	Linear regression	ANOVA	Linear regression	
	p-value	CorrCoef	R2	p-value	CorrCoef	R2	p-value	CorrCoef	R2	
Mean crown	0.366	0.185	0.03	0.002	0.645	0.42	0.000	0.867	0.75	
Height	
Mean dentin										
Cusp	0.618	0.103	0.01	0.000	0.817	0.67	0.000	0.738	0.51	
Mean dentin										
Fissure	0.300	0.212	0.05	0.056	0.424	0.18	0.308	0.092	0.01	
Mean										
Predentin cusp	0.762	0.062	0.00	0.910	0.026	0.00	0.641	0.042	0.00	
Mean										
Predentin fissure	0.787	0.056	0.00	0.646	0.107	0.01	0.000	0.504	0.25	
Mean height pulpal horn	0.829	0.045	0.00	0.420	0.186	0.04	0.000	0.416	0.17	
Notes.

CorrCoef Correlation coefficient

The ANOVA of the combined data with the real age of the animals as independent factor showed significant age effects for all variables, except for the mean dentin thickness in the fissure (p = 0.308), and the mean predentin thickness in the cups (p = 0.641) (Table 2). The mean crown height decreased significantly with the real age (Fig. 4A), as did the thickness of the dentin layer in cusps (Fig. 4B). Both variables showed a high explained variance (R2) of 0.745 and 0.506 respectively. Both other variables, showing a significant time dependency show far smaller explained variances, namely 0.247 for the mean predentin thickness in the fissure (Fig. 4C) and 0.172 for the mean height of the pulpal horn (Fig. 4D).

Figure 4 Scatter plots of the combined measurement data from the animals from the young and the old group.

Linear regression s lines and 95% CI are given.

Post-hoc analysis showed in general, a significant larger crown height in the young than in the old group. The thickness of the dentin layer on all cusps is significantly larger in the young group than in the old one, the predentin layer in the fissures is significantly thinner in the young group than in the old one, and finally the height of the pulpal horn in the young group is larger than in the old group (p-values for all these variables < 0.05).

Discussion

In many textbooks it is suggested that changes in the pulp tissue due to the aging process, such as reduction of the number and volume of the blood vessels or an increase in collagen fibers are diverse and irreversible (Hargreaves & Cohen, 2011; Hargreaves, Goodis & Seltzer, 2002). However, very little information is available about the additional influence of orthodontic force application on the aged pulp tissue. Therefore, the aim of the present study was to investigate the effect of experimental orthodontic tooth movement on young or aged dental pulp tissue. The rat was chosen as experimental animal and an appliance was used that has been developed earlier (Ren, Maltha & Kuijpers-Jagtman, 2004; Ren et al., 2003). This appliance has been proven to be stable and simple and is able to deliver a continuous and constant force as low as 10 cN on all three molars together. The total tooth movement over the 12 weeks experimental period was significantly more for the young than for the old animals. This force is comparable with a force of 170 cN on a human molar and thus more comparable to the clinical situation, than in most of the rat models (Abi-Ramia et al., 2010; Bletsa, Berggreen & Brudvik, 2006; Grunheid, Morbach & Zentner, 2007; Santamaria et al., 2007; Shigehara, Matsuzaka & Inoue, 2006; Tripuwabhrut et al., 2010) that use forces over 20 cN on only one molar (comparable to approx. 1,000 cN on human molar) (Ren, Maltha & Kuijpers-Jagtman, 2004). This might explain that in the present study only minor effects of orthodontic force application on pulpal tissues became apparent.

Another reason that no clear effects on pulpal tissues have been found in the present study might be that our appliance induced an almost bodily movement of the molar block (Ren, Maltha & Kuijpers-Jagtman, 2004; Ren et al., 2003), while others used tipping movements of the first molar only (Abi-Ramia et al., 2010; Bletsa, Berggreen & Brudvik, 2006; Grunheid, Morbach & Zentner, 2007; Santamaria et al., 2007; Shigehara, Matsuzaka & Inoue, 2006; Tripuwabhrut et al., 2010), intrusion (Barwick & Ramsay, 1996; Ikawa et al., 2001; Konno et al., 2007; Raiden et al., 1998; Ramazanzadeh et al., 2009; Stenvik, 1971; Veberiene et al., 2009), or extrusion (Ramazanzadeh et al., 2009; Stenvik, 1971; Subay et al., 2001).

The only pulpal change that was consistently found was a small increase in the vascularity in both the young and the old tooth movement group. This is in agreement with several other authors (Nixon et al., 1993; Raiden et al., 1998; Sano et al., 2002; Shigehara, Matsuzaka & Inoue, 2006; Wong et al., 1999). A drawback of the chosen measurement protocol in which the first histological evaluation is performed after one week is that short-lasting changes in the pulpal tissue, as described in the literature will be missed (Barwick & Ramsay, 1996; Bletsa, Berggreen & Brudvik, 2006; Brodin, Linge & Aars, 1996; Ikawa et al., 2001; Nixon et al., 1993; Santamaria et al., 2007; Santamaria et al., 2006). However, this is not a serious problem as short-lasting pulpal changes are of less clinical importance than persisting ones.

The finding that the dimensional variables in the young animals did not show any significant time effect can be explained in two ways. The first is that they do not change at all over the 12-week period; the other is that the changes are very slow and variable. As the changes in the dimensional variables are all related to occlusal wear of the dental tissues, it seems reasonable to suppose that their changes over time will be very slow as long as the crown is covered by enamel. During the 12-week experimental period in the old animals, a significant decrease with a high explained variance is found for the mean crown height and the mean thickness of the dentin in the cusps. This suggests a faster wear in the absence of occlusal enamel, which was probably worn by normal physiological processes before the animals were included in the study at an age of 40 weeks. As a consequence, secondary dentin deposition in the pulpal horns was stimulated, leading to a significant increase in its thickness, a decrease in the heights of the pulpal horns, and a decrease in the volume of the pulp chamber. The analysis of the combined data from the young and the old animals points in the same direction.

Conclusion

In conclusion, it can be stated that the application of a light force for orthodontic tooth movement in young nor adult rats did lead to long-lasting or irreversible changes in pulpal tissues, except for a tendency towards a higher vascularity. The most apparent changes are in the dimensional variables, but these are caused by occlusal wear and secondary dentin formation, and not by the orthodontic tooth movement. However, extrapolating this experimental data from an animal study in rats to human outcome is difficult and it is not surprising that severe wear as seen in rodents, is seldom encountered in adult orthodontic patients. Therefore, these findings have only very limited clinical implications.

Supplemental Information

Supplemental Information 1 Raw data file

Click here for additional data file.

Additional Information and Declarations

Competing Interests

Author Contributions

Animal Ethics

Data Availability

The authors declare there are no competing interests.

Martina Von Böhl and Jaap C. Maltha conceived and designed the experiments, performed the experiments, analyzed the data, wrote the paper, prepared figures and/or tables, reviewed drafts of the paper.

Yijin Ren and Piotr S. Fudalej analyzed the data, wrote the paper, reviewed drafts of the paper.

Anne M. Kuijpers-Jagtman conceived and designed the experiments, analyzed the data, wrote the paper, reviewed drafts of the paper.

The following information was supplied relating to ethical approvals (i.e., approving body and any reference numbers):

Board for Animal Experiments of the Radboud University Nijmegen—refernce number: KUNDEC 2001-07.

The following information was supplied regarding data availability:

The raw data has been supplied in the Supplemental Information.

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
