# Peer review of "Age-related changes of dental pulp tissue after experimental tooth movement in rats"

_PeerJ, doi:10.7717/peerj.1625_

## Round 0.1 · original submission · Major Revisions

Please address the several concerns noted by our reviewers.

·

Basic reporting

- Introduction: previous reports needed to be cited with caution. The authors stated the following context based on the previous reports (lines 83-84): it could be ‘concluded’ that orthodontic intervention on adolescents might not cause irreversible or long-lasting changes in pulpal physiology based (lines 84-86). Some of those previous studies cited were performed on rat models (Abi-Ramia et al. 2010, Santamaria et al. 2007), not on human subjects. Especially the second one (Santamaria et al. 2007) was performed on ‘adult’ animals, not on young ones. Therefore, it is not appropriate to cite these reports to mention that it could be ‘concluded’ that orthodontic intervention on adolescent human subjects did not cause irreversible or long-lasting changes in pulpal physiology (lines 84-86).

- Discussion: One of the pitfalls of the research design that the authors pointed out is that they retrieved their first data at 1 week after orthodontic tooth movement, and they suggested physiological changes that may occur within shorter period of time (< 1 week) might have missed. It is recommended to explain in discussion part what they could have done differently to achieve the information that they mentioned they might have ‘missed’. (for example, assessment on pro-inflammatory cytokines over short-term.)

- In the paragraph of the line 251, it is stated that an almost bodily tooth movement in this study might be one of the reasons why there was no clear effects on pulpal tissue.
o This statement is conflicted with the lines of 78-79, where the authors mentioned that temporal changes in pulp physiology such as blood flow and the number of blood vessels could result from bodily tooth movement as well as other types of orthodontic tooth movement.
o There is no valid data confirming that an almost bodily tooth movement was achieved in the experimental groups of this study.

- In the adult group, only the mean crown height significantly decreased and the mean thickness of the dentin in the cusps significantly increased (lines 216-217, Table 1) in the experimental groups. Significant changes in adult group between control and experimental groups are findings with the most interest in this study. However, these findings were not sufficiently interpreted anywhere in the manuscript. The authors may explain thoroughly what these findings mean.

- Unclear definition:
o Crown height (line 144):recommended to be rephrased as ‘crown height related to a line through the cementoenamel junction (CEJ)’
o The height of the pulpal horns, related to a line through the CEJ in the fissures (line 146-147):recommended to be rephrased as ‘The height of the pulpal horns, related to a line through the fissures’
--> CEJ is not related for this measurement (13, 14, 15)

- Minor typo (for example, line 44) and space (for line 147)

Experimental design

- Can the authors provide any supportive outcomes proving that there was actual orthodontic tooth movement achieved in the “experimental groups”?: Actual tooth movement by orthodontic intervention is an important and essential factor in this study, and the authors stated that the appliance and light force that they used in this study was known to be advantageous (line 126), simple, stable (line 244), and comparable to the clinical situation (line 246). However, there is no clear data in this manuscript that proved the molars on the experimental side were significantly moved by the orthodontic appliance.
o There is a chance that actual orthodontic tooth movement achieved in this study was too minimal to show changes in pulpal physiology. (for example, due to ankylosed molars, too heavy anchorage on the three molars as they were tied/combined together, or 10cN light force was not just sufficient to move the teeth) The authors also stated that they might have had minor effect of orthodontics force application in this study (lines 248-249).
o If the authors could present data showing actual tooth movement, it may contribute to support the hypothesis that orthodontic tooth movement did not affect pulpal physiology on both adolescent and adult animal models.
o Histological data that presents tension/compression sides of the roots/alveolar bones could be a reasonable data to elucidate actual tooth movement by the orthodontic appliance. (It is might not feasible to manually measure the magnitude of the tooth movement on small animal models such as rats.)

Validity of the findings

- Results of this study consist of two sets of data; histology (qualitative data) and dimensional parameters (quantitative data). For the qualitative data set, it is recommended to provide either all the raw measurement values or, if it is too much, their mean and standard deviation values besides the statistical summary shown in Table 1.
o This will help the readers to understand the authors’ findings. For example, context on lines 266-268 is not possible for the readers to follow with just the statistical analysis shown on Table 1. Additional data (e.g. raw data or at lease mean and standard deviation values) are required.

- The authors concluded that the only pulpal change observed from orthodontic tooth movement in both young and old groups was increased vascularity (lines 258-259). Specifically, it was stated that the number of blood vessels and their diameter increased after orthodontic tooth movement. (lines 180-181 and 199-200). The only data that supported this finding/conclusion is Fig. 3-b and Fig. 3-d (histology data). However, the histological windows shown there were so limited that it was wondered if they are representative enough to show increased number of the blood vessels. It was hard to agree the number of the blood vessels increased from the data (Fig. 3-b and Fig. 3-d). As the increased vascularity after orthodontic tooth movement is, if not the most important, one of the major findings of this study, it is recommended to have more solid data that could support the authors’ rationale than what they have included in this manuscript (Fig 3-b and d). Here are a few suggestions.
o A. Increased vascularity/angiogenesis can be supported by providing other genotypic/phenotypic assessments from additional experiments (most preferred one).
o B. More vivid data can be provided; a histology data with a larger area under lower magnification showing increased number of blood vessels more clearly. Quantitative data (i.e. number of blood vessels/area) may help supporting their interpretation.
o C. If it is not feasible to perform additional experiment (suggestion A), or gain further data (suggestion B), it can be thoroughly discussed as one of the limitations of this study or further studies (least preferred one).

Additional comments

As limited information has been reported regarding the effect of orthodontic force application on the aged pulp tissue, this study has a relevant and meaningful research rationale. Parts of the outcomes supported the changes in the pulpal physiology related to the aging process, which already has been well established. As the authors pointed out in lines 251-252, relatively less clear effects of orthodontic tooth movement on pulpal tissue physiology were elucidated in this study, which, otherwise, could have had higher impact. Although the manuscript has multiple limitations and shortcomings as listed above, I will be more than happy to re-evaluate it after major revision.

Reviewer 2 ·

Basic reporting

Authors of this study had tried to address an interesting clinical question, which is to study the effect of orthodontic tooth movement on pulpal changes in adults. They have utilized a rat model to test their hypothesis. Commendable aspect of this study is the usage of clinically relevant force levels to study the effect on pulp in adult rat teeth. However, this study has methodological concerns that need to be addressed.

Experimental design

1. It has been decided to use two groups of 30 male rats for control and experimental groups at 6 and 40 weeks of age for 5 time points (1, 2, 4, 8, and 12 weeks). This indicates that the study consisted of 3 animals per time point in each group. Authors have not reported power of this study; lack of adequate power could be a reason for not noticing significant differences between the group.

2. Histological sections of control and experimental groups were utilized for recording measurements. A major challenge in using histological section for measurement is to appropriately match the section from control and experimental group. Plane of embedding and sectioning are variables hard to control, could potentially increase the variability leading to lack of significance. Especially comparing the images from Fig. 3I and 3J, mesial root is present in the former and lacking in the later suggest that the position of the section is not well controlled.

3. It is not clear from reading the manuscript, how many sections from how many teeth were measured. For eg., did the authors measure 5 sections per molar?

4. Also it is not clear from the manuscript whether the authors confirmed any tooth movement actually occurred. In their model system as they have pitched 3 molars against 2 incisors, it is unlikely that the molars would have moved. It is important to confirm tooth movement in the experimental group compared to the control to ensure the effect of tooth movement related changes in adult rats.

5. Authors have chosen to study the coronal pulpal changes but not the radicular pulp. No convincing arguments had been presented in the manuscript for their choice of measuring coronal pulpal changes.

Validity of the findings

In light of questions raised regarding experimental design, making a meaningful conclusion of the results is challenging.

Additional comments

Authors effort in attempting to address an important clinical question is commendable. Addressing the questions regarding experimental design and methodology will greatly enhance the manuscript.

Reviewer 3 ·

Basic reporting

Review:
The authors have presented a study focusing on identifying pulpal changes that occur due to orthodontic tooth movement by comparing 6-week versus 40-week old rats. I have identified a few concerns that need to be addressed.


Please review the manuscript for grammatical errors

Experimental design

Please revise the methodology section to improve clarity.

If the stainless steel ligature was drilled through the roots of the maxillary incisors, is there a possibility of anchorage loss, i.e. retraction of incisors. Please clarify in the methods section.

Was a power analysis performed to determine sample size per group/time point?

If maxillary molars were subject to orthodontic movement, please re-orient figure 2 to depict the same.

Please state the amount of molar mesialization achieved in the young versus old group

Please state the rationale for choosing the specific timepoints. Is there a chance that you might have missed a significant pulpal change within the first week? Did the authors attempt at comparing other timepoints with 8-12 weeks?

In Figure 3, please include appropriate histological slides to depict Experimental side in young and old rats at 1000μm magnification.

Please state the clinical relevance of the results when treating orthodontic patients.

Validity of the findings

The authors have stated a generalized statement as their conclusion. It is suggested to tone down and/or specify that these findings based on the experimental model utilized in this study

---

## Round 0.2 · Minor Revisions

We appreciate the major changes made by the authors in the manuscript based on reviewer's comments with regard to introduction, methods, discussion and inclusion of new tables. But there is still a discrepancy between what was claimed of being addressed in the rebuttal document with regard to figures (2 and 3) and inclusion of power analysis and the lack thereof in the main manuscript document.

Looking forward to receiving your updated manuscript with the above items addressed.

---

## Round 0.3 · accepted · Accept

Authors addressed the concerns adequately.